# Atomic-precision control of plasmon-induced single-molecule switching in a metal–semiconductor nanojunction

Youngwook Park [1]✉, Ikutaro Hamada [2], Adnan Hammud [3],
Takashi Kumagai[4], Martin Wolf [1] & Akitoshi Shiotari [1]✉

Atomic-scale control of photochemistry facilitates extreme miniaturisation of optoelectronic devices. Localised surface plasmons, which provide strong confinement and enhancement of electromagnetic fields at the nanoscale, secure a route to achieve sub-nanoscale reaction control. Such local plasmon-induced photochemistry has been realised only in metallic structures so far. Here we demonstrate controlled plasmon-induced single-molecule switching of peryleneanhydride on a silicon surface. Using a plasmon-resonant tip in low-temperature scanning tunnelling microscopy, we can selectively induce the dissociation of the O–Si bonds between the molecule and surface, resulting in reversible switching between two configurations within the nanojunction. The switching rate can be controlled by changing the tip height with 0.1-Å precision. Furthermore, the plasmon-induced reactivity can be modified by chemical substitution within the molecule, suggesting the importance of atomic-level design for plasmon-driven optoelectronic devices. Thus, metal–single-molecule–semiconductor junctions may serve as a prominent controllable platform beyond conventional nano-optoelectronics.

Nano-optoelectronics represents a fascinating intersection of chemistry, physics, and engineering, focusing on the manipulation of optical and electrical properties at the nanoscale[1]. This field leverages the optoelectronic properties of individual molecules to develop nanoscale functional devices for a variety of applications, including sensors, light-emitting diodes and photovoltaic cells. As one elementary indispensable function, single-molecule photoswitching, whereby the geometry and/or electronic conductance of the molecular junction can be reversibly switched by optical stimuli, has been intensively studied[2–4]. Conventional molecular photoswitches, however, frequently rely on photochromic molecules, such as azobenzene derivatives[5] and spiropyran derivatives[6], in contact with electrodes. Exploring molecule-based devices beyond material limitations should enhance functionality and flexibility further.

Localised surface plasmon (LSP) plays a key role in creating optoelectronic devices without chromophores. LSP-resonant nanojunctions can boost photochemical reactions, in some cases enabling distinct reaction pathways that are inaccessible via far-field excitation[7,8]. Scanning tunnelling microscopy (STM) combined with laser excitation has allowed the characterisation of plasmon-induced reactions in real space[9–13], and recently achieved controlled photochemistry at the submolecular scale[14,15]. The LSP in the nanojunction and its control by STM would be utilised for accessing and controlling optoelectronic properties at the atomic scale.

Here we propose metal–molecule–semiconductor junctions as a versatile platform for single-molecule optoelectronics based on LSP-induced chemical reactions. Previous optoelectronic studies for metal–molecule–semiconductor nanojunctions have focused on the photovoltaic effect under far-field irradiation[16,17], but plasmon-mediated

[1]Department of Physical Chemistry, Fritz-Haber Institute of the Max-Planck Society, Berlin, Germany. [2]Department of Precision Engineering, Graduate School of Engineering, Osaka University, Suita, Japan. [3]Department of Inorganic Chemistry, Fritz-Haber Institute of the Max-Planck Society, Berlin, Germany. [4]Institute for Molecular Science, National Institutes of Natural Sciences, Okazaki, Japan. ✉e-mail: park@fhi-berlin.mpg.de; shiotari@fhi-berlin.mpg.de

chemical reactions on semiconductors have not been characterised. Unlike a metal−metal nanojunction, where a gap-mode plasmon with a large electromagnetic field can be excited[8], a nonplasmonic semiconductor substrate, such as a silicon (Si) surface, is expected to yield smaller field enhancement. Overcoming this challenge, given the substantial variety of functional molecule−semiconductor systems[18−25], expanding plasmon-driven processes to a semiconductor platform is highly motivating and promises miniaturisation of optoelectronic devices and acquisition of novel functions due to effective light−matter interactions[26].

As a demonstration, we show single-molecule switching of perylene-3,4,9,10-tetracarboxylic dianhydride (PTCDA) and its derivatives incorporated in the nanojunction consisting of a plasmonic Ag tip and a non-plasmonic Si(111)-7 × 7 reconstructed surface. The adsorption of PTCDA on the semiconductor surface has been well characterised[27,28] because of its potential interest for molecule-based electronics[29]. We employ a specifically designed low-temperature STM that enables a precise focusing of laser onto the tip apex and an efficient collection of tip-enhanced Raman scattering (TERS) signals[30]. For better control of the plasmon properties, we fabricate sharp Ag tips using a focused ion beam (FIB)[11,30]. When the

single-molecule junction is illuminated at the LSP-resonant wavelength, the dissociation of chemical bonds between Si atoms of the surface and acyl groups of the molecule is induced, leading to switching through the reversible formation/dissociation of an atomic point contact between the tip and the molecule (Fig. 1a). No intramolecular reaction occurs in PTCDA, unlike the conventional molecular photoswitches which operate through the reactions of photochromic moieties[5,6]; nevertheless, the reversible formation and dissociation of the molecule−electrode inter-actions allow the system to be classified as a single-molecule switch[31−37]. The plasmonic properties and three-dimensional positioning of the Ag tip determine the reactivity of the junction, enabling selective switching of the target moiety of the single molecule. Our demonstration of the atomic-scale photoswitching control possibly propels us into further miniaturisation of conventional nano-optoelectronics.

## Results and discussion
### Characterisation of PTCDA/Si(111) and molecular switching
Individual PTCDA molecules adsorb at the corner hole site of the Si(111)-7 × 7 surfaces, forming a bridging structure with four O−Si

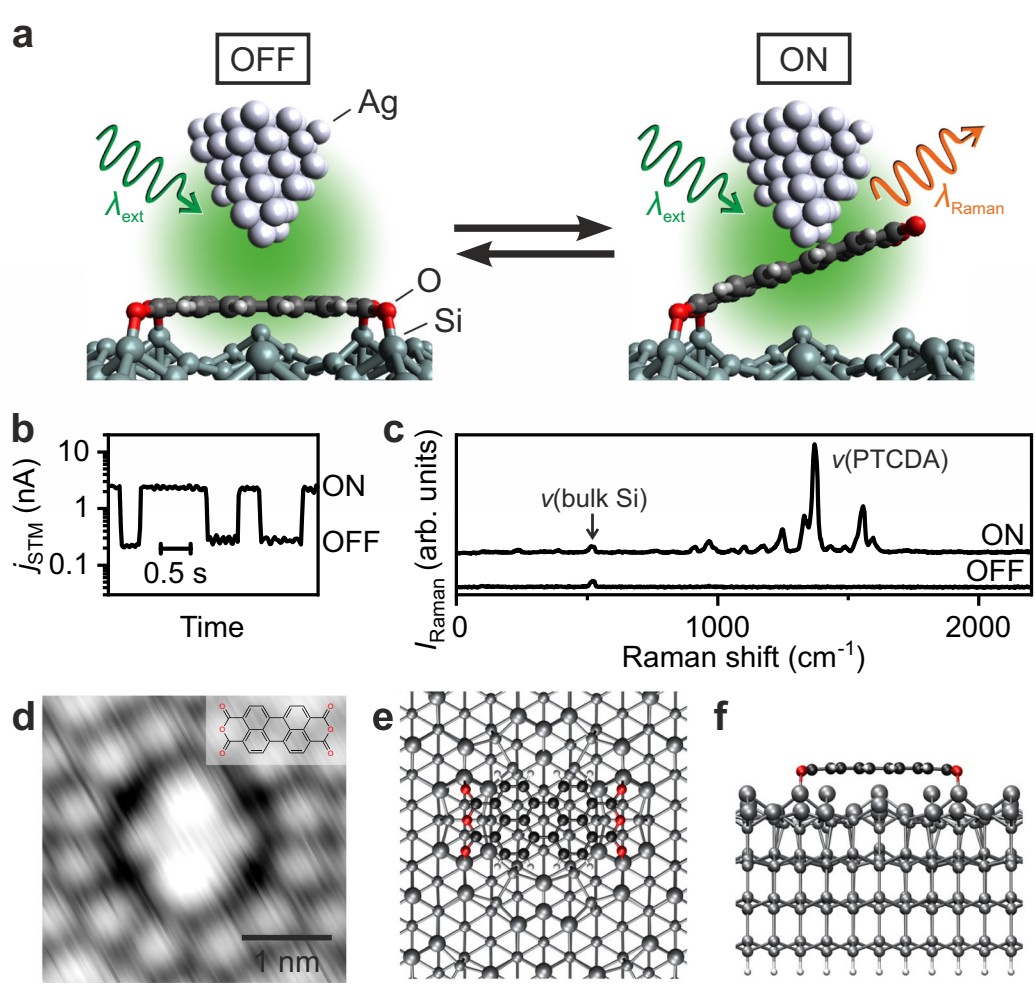

**Fig. 1 | Single-molecule switch driven by a localised plasmon. a** Schematic side-view of the plasmon-driven switching of a single PTCDA molecule adsorbed on a Si(111)-7 × 7 surface. The left (right) panel indicates the tunnelling junction (point-contact formation) between an Ag tip and PTCDA/Si(111), defined as the OFF (ON) state. $\lambda_{ext}$ and $\lambda_{Raman}$ denote the wavelengths of the incident laser illuminating the Ag tip and the Raman scattering from the STM junction, respectively. **b** Time trace of the STM current $j_{STM}$ during the switching. The current was recorded with an Ag tip located over a PTCDA adsorbate (gap distance between the tip and the OFF-state molecule $d = 1.8$ Å, sample bias $V_{bias} = -150$ mV, sample temperature $T = 78$ K) under visible-laser irradiation ($\lambda_{ext} = 532$ nm, incident laser power $P_{ext} = 0.56$ mW).

The low- (high-) current level corresponds to the OFF (ON) state. **c** TERS spectra of the OFF (bottom curve) and ON (top curve) states of PTCDA/Si(111) recorded with an Ag tip at $d = 2.5$ Å and 1.5 Å, respectively ($V_{bias} = -300$ mV, $T = 10$ K, $\lambda_{ext} = 532$ nm, $P_{ext} = 5.6$ mW). **d** STM image of PTCDA/Si(111) ($j_{STM} = 50$ pA, $V_{bias} = 500$ mV, $T = 78$ K). The inset shows the molecular structure. **e, f** Top and side views of the DFT-calculated structure of PTCDA/Si(111). The white, black, red and grey spheres represent H, C, O and Si atoms, respectively. For clarity, the Si atoms in the first and second layers are displayed as larger spheres (another display style is shown in Supplementary Fig. 1a).

bonds between the acyl O atoms and Si adatoms adjacent to the corner hole[27,28]. With STM at positive sample bias ($V_{bias} > 0$), the single molecule is imaged as a four-petal-shaped protrusion (Fig. 1d), where two darker (brighter) petals correspond to the long (short) side of the rectangular molecule[27]. Our density functional theory (DFT) calculations revealed that the molecule remains nearly flat (Fig. 1e, f), unlike a convex structure predicted previously[28] (see Supplementary Note 1 and Supplementary Fig. 1).

Plasmon-induced switching occurs when a plasmonic Ag tip under laser irradiation approaches the PTCDA molecule. The switching is manifested as a two-state telegraph noise of the STM current ($j_{STM}$) (Fig. 1b), indicating a reversible reaction of the molecule. Single-molecule junctions can be further characterised by vibrational spectroscopy recorded simultaneously with conductance measurements[38–41]. As shown in Fig. 1c (see Supplementary Note 2 and Supplementary Fig. 2 for measurement details), the intense Raman peaks originating from PTCDA are observed at the high conductance state (namely, ON state), while no molecular peaks appear at the low conductance (OFF) state. Note that the peak at 520 cm$^{-1}$ corresponds to the optical phonon mode of the bulk Si[42,43]. It has been previously shown that the TERS intensity is largely enhanced by the point-contact formation between a plasmonic tip and a single adatom or molecule on surfaces[42,44,45]. Therefore, we assign the high conductance state to a point-contact configuration, as depicted in Fig. 1a, where the O–Si bonds of one of the two anhydride groups of the molecule are broken and the dissociated half is lifted. The tip-height dependence of the junction conductance supports the assignment of the ON state geometry, as described in the next subsection.

## Tip-height dependence of the switching activity: 0.1-Å order control

The switching is activated in a particular range of tip–molecule gap distance ($d$) and we categorise $d$ ranges into Zones I (tip far) to III (tip close) (Fig. 2a), based on the characteristic behaviour of the $j_{STM}$ (Fig. 2b) and TERS spectra (Fig. 2d–h) recorded simultaneously. This measurement was conducted at 10 K to eliminate thermal drift effects, but the same Zone behaviours were observed at 78 K (Supplementary Fig. 3). The forward (tip approach) and backward (tip retraction) traces show the same tip-height dependence, indicating that the Ag–PCTDA–Si junction acts reversibly and the process was not destructive. Note that the switching event is not coincidental but well reproducible; once we found the proper tip conditions, we could repeatedly obtain switching features until the junction deformed (see also Supplementary Note 3 for details).

In Zones I and III, $j_{STM}$ increases exponentially as the tip approaches the molecule (blue ribbon in Fig. 2b), indicating that no configurational changes or chemical reactions are involved. In addition, the absence of the TERS peaks in these Zones (Fig. 2d, f, and h) suggests that the molecule remains in the OFF state. In contrast, in Zone II, $j_{STM}$ shows the telegraph noise (Fig. 2b) and the TERS peaks appear (Fig. 2e, g). In the high-conductance state (ON state), the measured current does not vary with the tip-sample distance (red ribbon in Fig. 2b). The invariant conductance across different tip heights indicates the formation of point contact between the tip and the molecule[44], consistent with the assignment by TERS described above.

The ON-state occupation derived from the telegraph noise of $j_{STM}$ shows sensitive $d$ dependence (purple dotted curve in Fig. 2c), which indicates that a change in tip height affects the relative stability between the OFF and ON states. In other words, the ratio between the forward and backward reaction rates can be adjusted by changes with 0.1-Å-order precision in $d$ (see also Supplementary Fig. 4). The $d$ dependence of the ON-state occupation is in good agreement with that of the TERS intensity (black solid curve in Fig. 2c). This is consistent with the ON- and OFF-state TERS spectra shown in Fig. 1c because only the ON state of the switching molecule during the signal accumulation

time (3 s) contributes to the TERS intensity. We also confirm that when the tip approaches further than Zone III and contacts the OFF-state molecule ($d = 0$; namely, Zone IV), the TERS signals of the OFF state are detected (Supplementary Fig. 3). The spectral features differ from those of the ON state, indicating the different molecular configurations between the two states (Supplementary Note 4).

The switching behaviour of PTCDA/Si(111) is independent of the lateral tip position (Supplementary Fig. 5), and it occurs both over the molecular centre (perylene part) and over the edge (anhydride part). We conclude, therefore, that in the ON state, the perylene part attractively interacts with the tip apex, analogous to the point-contact formation between a metal tip and a benzene ring reported previously[34] (see Supplementary Note 5 for a detailed discussion). Assuming that PTCDA has a rigid plane without intramolecular deformation, the tilting angle of the ON-state molecule (in Zone II) is estimated to be about 10–15° from the flat OFF configuration.

## Chemical dependence: active anhydride, silent imide

PTCDA has various derivatives with a similar molecular frame but different chemical properties[46,47]. They can be used to tune the switching property without a significant configurational change in the tip–molecule–substrate junction. In this work, we examined two derivatives, perylene-3,4,9,10-tetracarboxylic monoimide monoanhydride (PMI) and perylene-3,4,9,10-tetracarboxylic diimide (PDI). In the STM images at positive $V_{bias}$, PTCDA, PMI and PDI appear as a symmetric protrusion (Fig. 1d), a protrusion-and-depression pair (Fig. 3a) and a symmetric depression (Fig. 3i), respectively (see Supplementary Fig. 6 for more images at various $V_{bias}$). These appearances suggest that the brighter (darker) half of the image of PMI corresponds to an anhydride (imide) group (Fig. 3a, b).

To evaluate the switching behaviour, we recorded $j_{STM}$ at $d$ comparable to the active range of the PTCDA switching (Zone II). In contrast to PTCDA, switching of PMI shows a clear dependence on the lateral tip position (indicated by the markers in Fig. 3a, b). When the tip is placed over the anhydride side, the telegraph noise appears in $j_{STM}$ (Fig. 3e, f), whereas the tip over the imide side keeps $j_{STM}$ in a low conductance state (Fig. 3g, h). For PDI, no switching was observed at any tip locations (Fig. 3l). These results indicate that the anhydride side of PMI can be lifted (Fig. 3d) in the same manner as PTCDA, whereas PDI and the imide side of PMI do not react (Fig. 3c, k). The inertness of the imide groups strongly supports that the switching behaviour originates from the reactivity of the molecule–substrate system (Fig. 1a), ruling out the possibility of the switching due to the atomic-scale deformation of the plasmonic electrode (tip apex) in the picocavity reported previously[48]. For PMI, the separation between the switchable and non-switchable positions is 4 Å (orange and cyan in Fig. 3a, b), which highlights the importance of the sub-nanoscale tip positioning for the reaction control.

The different switching behaviours between the three molecules are explained by their adsorption energies $E_{ads}$. Our DFT calculations identify that PMI and PDI adsorb at the corner hole site of the surface via the four acyl O atoms in the same manner as PTCDA (Supplementary Fig. 1). $E_{ads}$ is calculated to be −4.21 eV, −4.41 eV, and −4.85 eV for PTCDA, PMI and PDI, respectively, which indicates the stronger interaction of the imide group with the Si surface than the anhydride group.

In terms of junction sustainability, PMI has an advantage over PTCDA. Although non-destructive switching is feasible for both molecules (Fig. 2), lateral diffusion or pick-up of PTCDA by the tip was occasionally observed in Zone II or at smaller tip heights (cf. $d \approx 6$ Å for STM imaging vs $d \approx 2$ Å for switching). In contrast, such irreversible events were rarely observed for PMI because the non-reactive imide side acts as a stable anchor bound to the surface. This provides an important insight into designing stable single-molecule switching in nanojunctions.

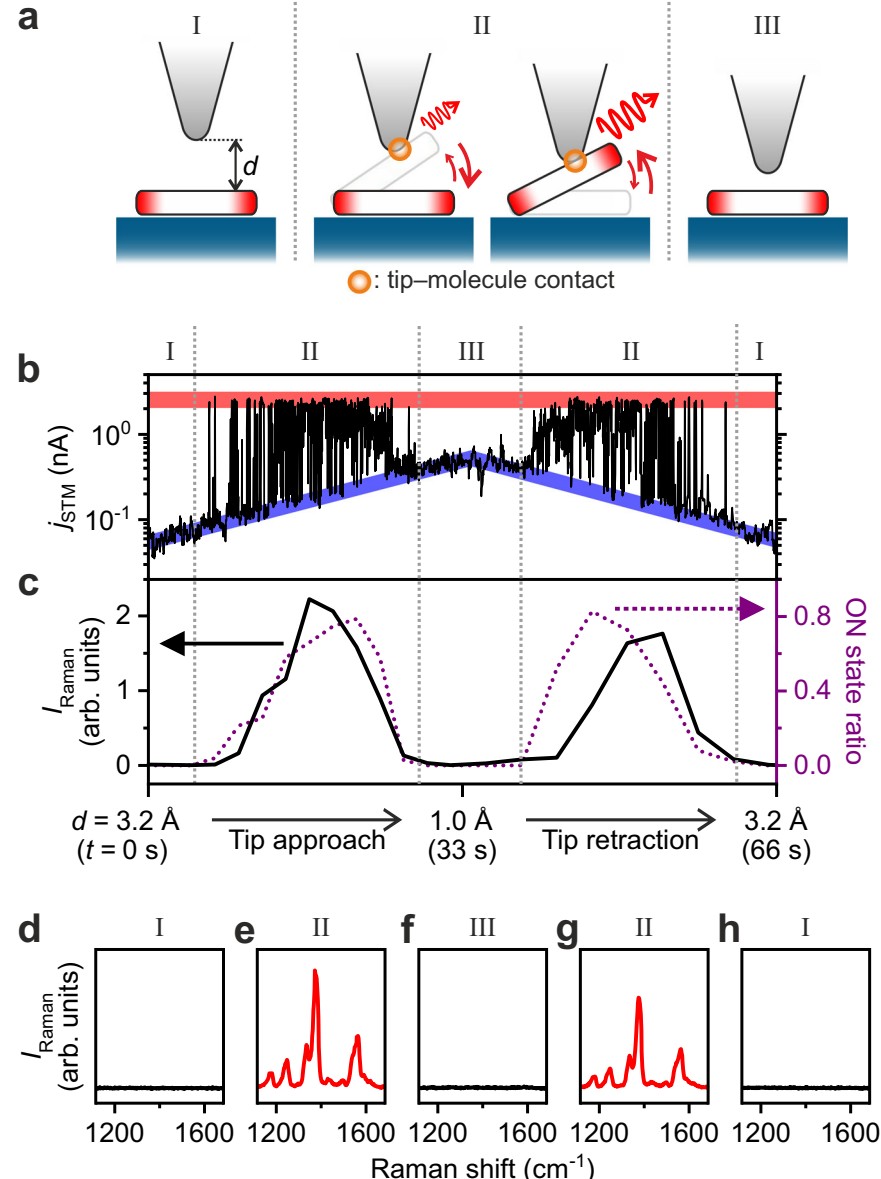

**Fig. 2 | Activation and control of the molecular switch by tip-height tuning.**
**a** Schematics of the Ag–PTCDA–Si(111) junction at four different gap distances:
from left (tip far) to right (tip close), Zone I, Zone II with low ON-state occupation,
Zone II with high ON-state occupation and Zone III. **b** Time trace of $j_{STM}$ during the
approach-and-retraction stroke of an Ag tip over PTCDA/Si(111) (20-ms time reso-
lution, $V_{bias} = -400$ mV, $T = 10$ K, $\lambda_{ext} = 532$ nm, $P_{ext} = 5.6$ mW). The first (last) half of
the record time, i.e. $t = 0$–33 s (33–66 s), is the data during the tip approach

(retraction) process, where $d$ was varied by $-0.2$ ($+0.2$) Å every 3 s. Red (blue)
ribbons indicate guides for the high (low) current level corresponding to the ON
(OFF) states. **c** TERS intensity of a 1375-cm⁻¹ peak measured simultaneously with
$j_{STM}$ in (**b**). The purple dotted curve shows the ON-state occupation ratio calculated
from the $j_{STM}$ trace at each $d$. **d**–**h** TERS spectra in individual Zones acquired during
the trace in (**b**). The spectra in **d**–**f** (**g**, **h**) were recorded during the tip approach
(retraction) process (from **d** to **h**, $d = 3.2$ Å, 2.1 Å, 1.0 Å, 2.1 Å and 3.2 Å).

## Reaction mechanism

The results above suggest that switching of the PTCDA and PMI
molecules on the Si surface (referred to as the anhydride/Si switch) is
mediated by LSP in the Ag-Si nanojunction. However, other stimuli,
such as $V_{bias}$-derived tunnelling electrons, electrostatic field or
tip–sample attractive/repulsive forces, inevitably coexist in the STM
junction, all of which could also contribute to the molecular
reaction[21,49]. We conducted the following control experiments to
identify the driving force of the O–Si dissociation to activate the
anhydride/Si switch.

First, we confirm that light irradiation to the STM junction is
necessary for the switch activation (Fig. 4a and see Supplementary
Fig. 7 for the laser-power dependence); the telegraph noise in $j_{STM}$
disappeared in the absence of laser irradiation independent of $V_{bias}$.

Second, by modifying the plasmonic resonance of the junction
through the tip-apex shaping (see "Method"), we verify the near-
field contribution to switching. We prepared Tips 1 and 2 and con-
firmed their different plasmon-resonance energy profiles (Fig. 4b)
by STM-induced luminescence (STML)[50]. The resonance at around
500 (750) nm for Tip 1 (2) suggests that the LSP is resonantly excited
by incident light with a wavelength $\lambda_{ext}$ of 532 (780) nm (see the
markers c and e in Fig. 4b). Under the on-resonance conditions, the
switching was detected (Fig. 4c, e). In contrast, when 780-nm laser
irradiates Tip 1 (i.e. off-resonance), no switching was observed
(Fig. 4d). We also confirmed that an LSP-resonant Ag tip with
$\lambda_{ext} = 633$ nm induced the switching (Supplementary Fig. 8). We
conclude, therefore, that the anhydride/Si switch is initiated by
plasmon excitation.

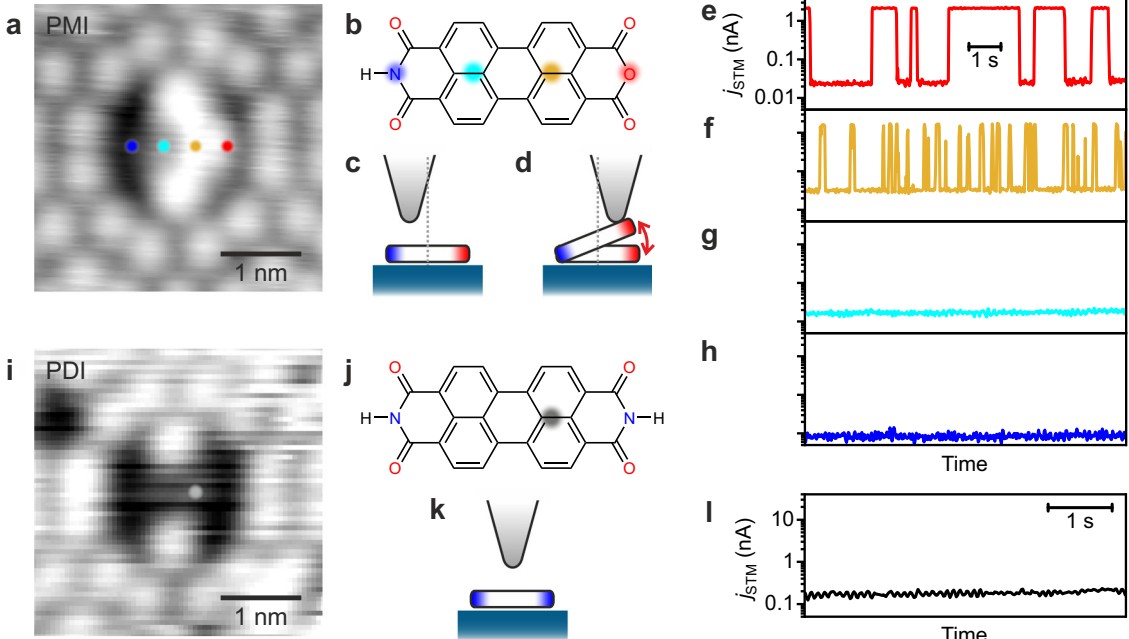

**Fig. 3 | Chemical tailoring of the switch. a** STM image of a PMI molecule on Si(111)-7 × 7 ($V_{bias}$ = 800 mV). **b** Structure of PMI. **c, d** Side-view schematics of the Zone-II junctions for PMI with different lateral tip positions: over the imide side (**c**) vs the anhydride side (**d**). **e–h** $j_{STM}$ traces recorded at different lateral tip positions over PMI/Si(111) ($V_{bias}$ = −300 mV, $\lambda_{ext}$ = 532 nm, $P_{ext}$ = 8 μW, $d$ = 1.5 Å). The four panels have identical scale axes. The coloured markers in **a** and **b** correspond to the tip location for the current traces with the same colours. **i** STM image of a PDI molecule on Si(111)-7 × 7 ($V_{bias}$ = 500 mV). **j** Structure of PDI. **k** Side-view schematic of the PDI junction. **l** $j_{STM}$ trace recorded over PDI at the position marked with the grey circles in (**i, j**) ($V_{bias}$ = 500 mV, $\lambda_{ext}$ = 532 nm, $P_{ext}$ = 5.6 mW, $d$ = 1.5 Å). In the figure, all measurements were conducted at $T$ = 78 K.

Plasmon-induced reactions can be triggered not only by direct transitions due to optical absorption of the reactant but also by hot carriers (HCs; high-energy nonequilibrium electrons and holes) and heat generated in the relaxation process of LSP[8,51–53]. In our case, the linear laser-power dependence of the switching rate (Supplementary Fig. 7) rules out the plasmon-mediated heat- or electric field-driven mechanism where nonlinear dependence is expected (see Supplementary Note 6 for a detailed discussion). The fact that the switching is observable with several $\lambda_{ext}$ also excludes the contribution of a mechanism based on direct optical excitation[10,15]. The described features resemble the HC-mediated dissociation of oxygen molecules strongly bound to an Ag surface in a plasmonic metal junction[12,13]. Therefore, we suggest that the anhydride/Si switch follows the HC transfer mechanism[54] (Supplementary Note 6 and Supplementary Fig. 7). In this mechanism, the reaction is triggered by HCs excited in the plasmonic electrode and transferred to the molecule, analogous to the Antoniewicz model established for electron-stimulated desorption[55]. The system gains sufficient kinetic energy to overcome the activation barrier by the excitation from the ground state (GS) to a transient charged state (CS) and its relaxation due to the different equilibrium internuclear distances between the two states (Fig. 4f).

The HC-driven mechanism has widely been proposed for plasmon-induced chemistry in metal−molecule−metal nanojunctions[9,12–14]. However, we emphasise that the Ag−molecule−Si nanojunction is distinct from the conventional systems. Our finite element method (FEM) simulations confirmed that, unlike an Ag surface, a Si substrate does not contribute effectively to plasmonic field enhancement in the STM junction (Supplementary Fig. 9). Consequently, HCs are dominantly provided from the plasmonic Ag tip, which probably contributes to the high controllability and target-selectivity of the switching by tip positioning (Figs. 2 and 3e–h). This contrasts with the metal−metal junctions, where the strong plasmonic electric field spreads spatially (>10 nm); HCs from the metal substrate can contribute to unspecific reactions of multiple molecules distributed over the surface area[9,10].

$V_{bias}$ independence of the switching behaviour (Fig. 4a) excludes the contribution of $V_{bias}$-derived tunnelling electrons[33,56] to the reaction. Theoretically, high-energy tunnelling electrons injected into the molecule by high $V_{bias}$ potentially induce the same reactions as plasmon-mediated HCs[9,12–14]. Nevertheless, we have not observed $V_{bias}$-induced switching of anhydride/Si even at a few volts (Fig. 4a; 3 V is the highest $V_{bias}$ that could be used for the $j_{STM}$ trace in Zone II non-destructively). Instead, the tip-apex structure tends to be irreparably destroyed at such high voltages probably because of an excessive electric field in the junction[57]. Based on the FEM simulations, the plasmon-derived electric field is estimated to be less than $2.3 \times 10^8$ V/m, while $V_{bias}$ of a few hundred mV will lead to an even higher direct-current field in the junction (Supplementary Note 7). This strongly suggests that the LSP-resonant tip acts as a supplier of high-energy electrons/holes (HCs) to invoke single-molecule reactions efficiently and non-destructively.

The short duration of the ON state restricts the obtainable information compared to the OFF state and impedes in-depth analysis of the ON-to-OFF reaction mechanism; nevertheless, there is considerable promise for addressing this through further chemical tailoring of the target molecule. Since the forward and backward reactions proceed on a comparable time scale (Fig. 1b and Supplementary Fig. 4), we predict that the HC-transfer mechanism is also responsible for the ON-to-OFF reaction. The ON-state occupation (Fig. 2c) is affected by the double-well potential shape of the GS (Fig. 4f). The 0.1-Å scale displacement of the tip height varies the ON-state potential energy and deforms the reaction barrier, thereby modifying the reaction rate.

We characterised single-molecule nanojunctions between an Ag tip and a Si surface using PTCDA and its derivatives, PMI and PDI, and controlled their plasmon-induced photoreactivity. Approaching the LSP-resonant tip under illumination to PTCDA/Si(111) induces the O−Si bond breaking to form a point-contact junction between the molecule

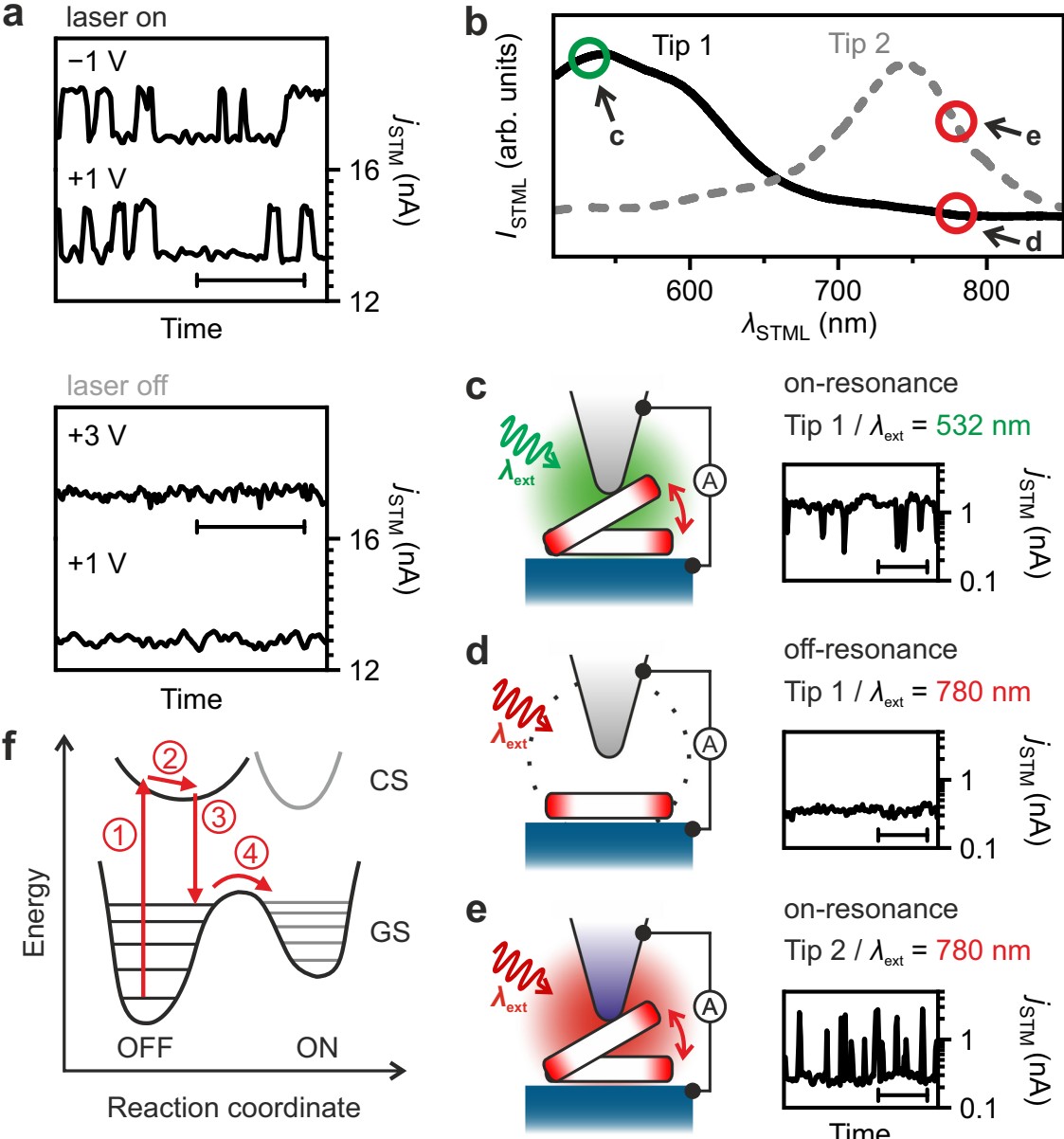

**Fig. 4 | Characterising plasmons as the driving force of the reaction. a** $j_{STM}$ traces for PTCDA/Si(111) with (upper panel) and without (lower) laser irradiation (laser on: $\lambda_{ext}$ = 532 nm, $P_{ext}$ = 56 μW, $d$ = 2.0 Å; laser off: $P_{ext}$ = 0, $d$ = 1.9 Å and 2.0 Å for traces at +3 V and +1 V, respectively). The voltage value above each trace denotes $V_{bias}$ used. The upper trace in each panel (−1 V and +3 V) is displayed in arbitrary units and offset vertically for clarity. **b** Normalised STML spectra recorded on a bare Si(111) surface with two Ag tips, namely, Tip 1 and Tip 2 ($j_{STM}$ = 10 nA and 40 nA, respectively, $V_{bias}$ = 3 V). The circles in the spectra indicate the STML intensities at the same wavelength as the irradiating laser used for the $j_{STM}$ trace measurements in (**c**–**e**). **c** $j_{STM}$ trace measured over PTCDA using Tip 1 at

$\lambda_{ext}$ = 532 nm ($V_{bias}$ = −300 mV, $P_{ex}$ = 5.6 mW, $d$ = 2.0 Å). **d**, **e** $j_{STM}$ traces recorded over PTCDA using Tips 1 and 2, respectively, at $\lambda_{ext}$ = 780 nm ($V_{bias}$ = −300 mV, $P_{ext}$ = 35.5 mW, $d$ = 2.0 Å). The junction schematics are displayed on the left side of each trace in (**c**–**e**). In the figure all measurements were conducted at $T$ = 78 K. All the horizontal scale bars in the traces denote 1 s. **f** Schematic potential energy diagram of the anhydride/Si switch under the plasmon-driven HC transfer. Red arrows indicate sequential steps: the transition from the ground state (GS) to a transient charged state (CS) ①, gain of kinetic energy due to the different equilibrium nuclear configurations between GS and CS ②, relaxation to a vibrational excited state in GS ③ and transfer across the barrier ④.

and the tip. The Si surface leads to strong chemisorption of the anhydride molecule via O−Si bonds and reduces electric field enhancement, which hinders undesired reactions or junction destruction. The LSP also induces the backward reaction, i.e. the dissociation of the contact point between the tip and molecule, leading to the switching behaviour. The switch operation requires both an effective plasmon resonance and an optimum tip−molecule gap distance, allowing for unprecedented control of the reaction rate by changing the gap distance with 0.1-Å precision. Moreover, by

comparing the three target molecules, we reveal that an imide moiety on the surface is inert against the plasmon in contrast to the reactive anhydride, although they have little difference in adsorption geometries. Such pinpoint chemical tailoring promises advanced design of single-molecule devices. The design and control of metal−molecule−semiconductor junctions should pave the way for the development of plasmon chemistry with high selectivity and controllability beyond conventional nano-optoelectronics, towards pico-optoelectronics.

## Methods

### STM and TERS experiments

The experiments were performed in an ultrahigh vacuum (UHV) chamber equipped with a low-temperature STM (modified UNISOKU USM-1400) at sample temperatures $T = 10$ K and 78 K. No difference between the two temperatures was found in the STM image appearances, $j_{STM}$ traces and TERS spectra of the molecules on the Si surface. The sample stage is equipped with three-dimensional coarse piezo motors and a piezo tube scanner whereas the tip is fixed at the STM unit. The bias voltage ($V_{bias}$) was applied to the sample while the tip was grounded. The STM images were obtained in the constant current mode at $j_{STM} = 50$ pA.

For the $j_{STM}$ trace and TERS measurements, we set the tip position as follows. First, the lateral tip position was set at the centre of the target molecule based on its STM image, and then the current feedback loop was opened. Next, $V_{bias}$ was set and the tip was vertically (and laterally, as needed) moved to a specific sampling point. This process guarantees an identical tip height for the lateral-tip-position dependent measurements (e.g. Fig. 3). The tip approaching procedure was conducted over the molecular centre, otherwise specified. The origin of the gap distance between the tip and the OFF-state molecule $d$ is determined by the point when the approaching tip first contacts the OFF-state molecule (i.e. the boundary between Zones III and IV) determined by the $j_{STM}$–$d$ curve (Supplementary Fig. 3).

As continuous-wave visible laser sources, 532- and 780-nm solid-state lasers and 633-nm HeNe laser were used. A $p$-polarised incident laser entered the UHV chamber through the corresponding laser-line filter and a fused silica window. The laser beam was precisely focused at the STM tip apex by an Ag-coated parabolic mirror (focal length of 8 mm, numerical aperture of 0.6) mounted on a five-axis ($x$, $y$, $z$, $\theta$ and $\phi$) movable piezo stage in the STM unit. The optimal focus was confirmed by the plasmon-induced downshift of a field emission resonance (FER) spectrum on a clean Ag(111) surface[58].

For TERS and STML measurements, the light scattered from the junction was collected by the parabolic mirror and then guided to a grating spectrometer (AndorShamrock 303i) outside the UHV chamber via a beam splitter. A long-pass filter with the cut-off wavelength corresponding to each incident laser was used for TERS. The acquisition time per TERS spectrum is 3 s. The STML spectra were recorded in the constant current mode without laser irradiation.

### Sample and tip preparation

Si(111) sample plates (Siegert Wafer GmbH; As doped) were degassed at 1023 K and then flash-annealed at 1473 K multiple times in the UHV chamber. The surface cleanness and $7 \times 7$-reconstruction formation were confirmed with the STM image. The molecules were loaded on a Knudsen-cell evaporator and degassed thoroughly under the UHV conditions. PTCDA (Tokyo Chemical Industry Co., Ltd.), PMI (synthesised as reported in the Supporting Information of ref. 47) and PDI (Tokyo Chemical Industry Co., Ltd.) were evaporated at 578 K, 598 K and 568 K respectively, with a clean Si(111)-$7 \times 7$ surface at room temperature facing the cell.

Chemically etched Ag wires were processed by Ga-ion FIB milling (see Supplementary Fig. 10) to obtain sharpened Ag tips with highly reproducible plasmonic property[30]. The Ag(111) surface for use in the FER measurement with the laser-irradiated Ag tip was prepared by Ar-sputtering-and-annealing cleaning cycles in the chamber. To change the plasmon-resonance energy profiles of the Ag tip (see Fig. 4b), we also used the clean Ag sample for the tip-apex adjustment, for which the tip was mildly poked into the surface and voltage pulses of a few volts for 0.1 s were applied in the Ag–Ag junction.

### Theoretical calculations

All the periodic DFT calculations were performed using the STATE code[59]. The exchange-correlation energy and potential were described using the rev-vdW-DF2[60] flavour of the van der Waals density functional (vdW-DF)[61]. The efficient implementation[62,63] of the self-consistent vdW-DF was used[64]. Pseudopotentials were used to describe the electron–ion interaction; the ultrasoft pseudopotential scheme[65] was used to describe the H 1$s$, C 2$p$, N 2$p$ and O 2$p$ channels, while the norm-conserving pseudopotential scheme[66] was used for other channels. The pseudopotentials were generated using the Perdew–Burke–Ernzerhof (PBE)[67] generalised gradient approximation functional, and the use of the PBE pseudopotentials was validated in ref. 68. Wave functions and augmentation charge were expanded in a plane-wave basis set with the kinetic energy cutoffs of 36 Ry and 400 Ry, respectively. The Si(111)-$7 \times 7$ surface was modelled using a slab consisting of eight Si layers (four Si bilayers) and a vacuum equivalent to twelve Si layers. The dangling bonds of the bottommost Si layer were terminated by H atoms. The rev-vdW-DF2 optimised lattice constant of 5.461 Å was used to construct the slab. Only the $\Gamma$-point was used to sample the surface Brillouin zone. A PTCDA, PMI or PDI molecule was put on one side of the surface, and the effective screening medium method[69,70] was used to eliminate the artificial electrostatic interaction with the image slabs. All the atoms were relaxed until the forces acting on them were less than $2.5 \times 10^{-2}$ eV/Å ($1 \times 10^{-3}$ Hartree/Bohr), except for the Si atoms in the bottommost layer and their terminating H atoms. The adsorption energy is defined by

$$E_{ads} = E_{tot}(\text{molecule/Si}) - E_{tot}(\text{molecule}) - E_{tot}(\text{Si}),$$

where $E_{tot}(\text{molecule/Si})$, $E_{tot}(\text{molecule})$ and $E_{tot}(\text{Si})$ are the total energies of the adsorption system, isolated (gas-phase) molecule and Si slab, respectively.

To evaluate the plasmon-enhanced electric field in the Ag–Si nanojunction, we also performed FEM simulations. The detailed method is described in Supplementary Note 7.

## Data availability

The source data of the figures in the main text and Supplementary Information in this study have been deposited in the Zenodo database under accession code https://doi.org/10.5281/zenodo.10890118. Other data supporting the findings of this study are included in this paper and the Supplementary Information.

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

## Acknowledgements

The authors thank Egbert Willem Meijer for providing the PMI molecule and Melanie Müller for the fruitful discussion. I.H. acknowledges the support of JSPS KAKENHI grant no. JP20H05660. T.K. acknowledges the support of JST FOREST grant no. JPMJFR201J and JSPS KAKENHI grant no. JP19K24684.

## Author contributions

A.S. conceived and directed the project. Y.P. and A.S. designed the experiments. Y.P. performed the STM-TERS experiments and analysed the data. I.H. conducted the DFT calculations. A.S. conducted the FEM simulations. A.H. fabricated the FIB-sharpened tips. Y.P., T.K., M.W. and A.S. examined the reaction mechanism. Y.P. wrote the original draft and T.K., M.W. and A.S. reviewed and edited the manuscript.

## Funding

## Competing interests

The authors declare no competing interests.
