## [Peer Review File · Nature Communications]

Atomic-Precision Control of Plasmon-Induced Single-Molecule Switching in a Metal-Semiconductor NanojunctionREVIEWER COMMENTS

Reviewer #1 (Remarks to the Author):

The authors realized controlled plasma-induced single-molecule switching of individual PCTDA molecules on silicon wafers based on plasmon-resonant tip in low-temperature scanning tunneling microscopy. They can selectively induce the dissociation of the O-Si bond between the PCTDA molecule and the surface, leading to reversible switching, and the switching rate can be controlled by the tip. Chemical tailoring of the target molecule also affects the properties of the switch.

This work differs from conventional single-molecule junction photo-switching in that it is a metal-molecule-semiconductor junction and the photo-switching is caused by plasmonic resonance acting on the molecule rather than by the molecule's own photo-excitation or photo-responsiveness, which has been less frequently reported previously. The authors show the change in STM current during tip movement, and the switching behavior is characterized by bistatic telegraphic noise. The STM imaging is very convincing for target molecules with different chemical modifications. Acquisition of tip-enhanced Raman scattering spectra is also performed simultaneously, and the Raman spectral intensity characterizes the strength of the plasmon resonance.

The study presented is interesting and I believe this manuscript is deserved to be published by Nature Communication finally. However, at the current state, I am wonder that the evidence confirming the mechanism of the switch is not convinced. The number of demonstrated current-time traces is limited and the sample ratio is not presented, i.e., it is not clarified whether it is an accidental behavior or a general reproducible phenomenon for the target molecule at the specific position.

Here are some detailed comments for the manuscripts:

1. I am wonder why the current show the telegram noise behavior when the Ag tip approaches the molecule. If the telegram noise behavior is due to the breaking and reforming of O-Si bonds at one end of the molecule upon SPP, then O-Si bonds at another end of the molecule can also break upon SPP, which will result in a three-states of current rather than two-level of current. Additionally, I am wonder why the O-Si bonds can reform frequently. If the break of O-Si is induced by the SPP enhanced vibration (Science, 2018, 360, 521), then, it will be great challenge to reform O-Si bonds frequently under a continuous resonant light illumination.
2. Please repeat the collection of current traces for this molecule at different locations to support the conclusion. Whether the switching behavior occurs every time and or once in a while. what is the percentage to observe such a behavior and what is the switching ratio. The above mentioned data can be statistically analyzed.
3. Whether the molecule under test has light absorption at the laser wavelength used, it is recommended to complement the UV-Vis spectrum to exclude this factor.
4. In Supplementary Note 2, it gives the reason for the decrease in TERS intensity as the tip gets closer to the surface in the on-state. It may possible due to the change of different plasmonic modes in the sub-nanometer quantum tunneling regime, from the BDP mode (charge transfer quenches the dipolar bonding plasmon (BDP), leading to decreasing plasmonic enhancement with the decreasing gap) to the CTP mode (the intensity of charge transfer plasmon (CTP) modes increases with the decreasing gap).
5. During the movement of the tip, the on or off state of the current appears to be random rather than a regular telegram behavior (Fig 2b). Can the authors really achieve controllable reversibility of the single-molecule switching ? Does this mean that the breaking and forming of O-Si bonds is a random process?
6. The authors have performed laser power dependence tests (Figure S7). It shows that the switching ratio is positively dependent on the laser power. While, this observation seems to conflict with the conclusions. When the laser power increase, the break of O-Si will become easy while the reformation of O-Si will become difficult simultaneously due to the asymmetric energy landscape upon SPP (Sci. Adv. 2022, 8, eabp9285).
7. Is it possible that the switching behavior is due to the action of optical gradient forces or the interaction between picocavity adatom dipole and the induced molecular dipole, which makes the bonds break or makes the tip atoms displacements (refer to Sci. Adv. 2022, 8, eabp9285).
8. For COMSOL based calculations, the distance of the gap is on the sub-nanometer scale, whether the finite element method is still applicable in this case and whether additional quantum effects need to be taken into account.
9. For discussion, can imide form N-Si bond on the substrate? Both N and O are electronegative, can other atoms be substituted here, such as C?

10. The authors mentioned "Conventional molecular photoswitches, however, frequently rely on photochromic molecules, such as azobenzenes and piropyrans", here, it is better to call it "azobenzene derivatives and piropyran derivatives" (J. Am. Chem. Soc. 2024, 146, 6856). Additionally, it is better to cite a newest review paper to give a comprehensive story for the auditors, e.g., Toward Practical Single-Molecule/Atom Switches (Adv. Sci. 2024, 2400877).

Reviewer #2 (Remarks to the Author):

I am happy to recommend publication of this manuscript in its current form. The authors provide convincing data for the single molecule switching of PTCDA and its derivatives which is induced by the plasmonic field in the STM nanogap with a silver tip and a silicon surface. The STM imaging and associated STM and spectroscopic data is of good quality. Control experiments have been carefully conducted to strongly prove the hypothesis. The switching is well mapped out through the 3 different height zones, I, II and III and simultaneously recorded TERS spectra record unequivocal evidence. The 3 molecules compared (PTCDA, PMI and PDI) and the associated position resolved STM data provide an excellent comparison, and the role of the adsorption energy is well clarified. I also like the control experiments with the two different STM tip shapes with different plasmonic responses. These and the other control experiments help to make a clear and exacting assignment to plasmonic phenomena. This is an important new contribution to the field of nano-plasmonics and single molecule plasmonic phenomena.

Reviewer #3 (Remarks to the Author):

The ms reports measurements of the tunnel current and TERS of PTCDA derivatives on silicon using a silver tip under irradiation. The authors observe two level fluctuations of the current as a function of the tip-substrate distance that they interpret as switching, i.e. uplifting the molecule to form a point contact with the tip. The authors investigate three different species with varying coupling strength to the substrate/tip and different positions of the tip on the molecule. Further control experiments are performed to support that plasmonic properties of the tip are instrumental for the observations. From the set of data the authors conclude that hot carriers mediate the switching. The data are in parts supported by DFT calculations. The results are novel, interesting and in general well explained and should be published. I have a few comments though:

1. It is really necessary to use the term pico-optoelectronics? I find this unnecessary and a bit overselling. This report is by far not the first investigating light-induced molecular instabilities. The new aspect here is that it is on a semiconductor surface, but this does not justify to invent a new name for it. In addition the molecules are not particularly small, they are several nm large. The only picoscale is the lift height, which is again a typical height scale in STM business. Furthermore this term does not do anything to the science discussed here. I suggest to remove this term.
2. The authors should define what they call a molecular switch. They use a wider definition than in many other works. In the present ms nothing switches inside the molecule. It is the contact to the tip that switches which is not an intrinsic property of the molecule under investigation.
3. I find the quoted references not really adequately chosen. Several groups have investigated molecules on semiconductor surfaces and also "switching" behavior without always using this term, e.g. the Grill group, the Ernstorfer group and several others. Their work should also be referred to. Instead the authors refer extensively to their own work in a summarizing manner (e.g. refs. 22-25), where a reference to one single work would have been sufficient.
4. In line 168 the authors write: "Plasmon-induced switching occurs when a plasmonic Ag tip approaches the surface...". The authors should add the information that this happens under irradiation and not just by approaching. The term "plasmonic tip" alone is not sufficiently well established to understand that this happens under irradiation only.
5. In line 217 the authors refer to quantized conductance to be seen in Fig. 2b. I do not see any quantized conductance there, nowhere in this work. Also the reference [25] is not an adequate reference for quantized conductance. A quantized conductance would give rise to currents in the range of 30 μ A for 400 mV. The current here is in the nA range. Probably the authors mean discrete or saturated values of the conductance, but even these are hard to detect and that Fig. In Fig 1b one sees constant current over a certain period of time, but also no quantization. A constant current may also signal saturation of the amplifier. This issue must be resolved.

6. It would also be fair to refer to other works in which simultaneously single-molecule Raman and conductance have been studied, e.g. by the Natelson group and the Barth group.

Reply to Comments of Reviewer 1

The authors realized controlled plasma-induced single-molecule switching of individual PCTDA molecules on silicon wafers based on plasmon-resonant tip in low-temperature scanning tunneling microscopy. They can selectively induce the dissociation of the O-Si bond between the PCTDA molecule and the surface, leading to reversible switching, and the switching rate can be controlled by the tip. Chemical tailoring of the target molecule also affects the properties of the switch.

This work differs from conventional single-molecule junction photo-switching in that it is a metal-molecule-semiconductor junction and the photo-switching is caused by plasmonic resonance acting on the molecule rather than by the molecule's own photo-excitation or photo-responsiveness, which has been less frequently reported previously. The authors show the change in STM current during tip movement, and the switching behavior is characterized by bistatic telegraphic noise. The STM imaging is very convincing for target molecules with different chemical modifications. Acquisition of tip-enhanced Raman scattering spectra is also performed simultaneously, and the Raman spectral intensity characterizes the strength of the plasmon resonance.

The study presented is interesting and I believe this manuscript is deserved to be published by Nature Communication finally. However, at the current state, I am wonder that the evidence confirming the mechanism of the switch is not convinced. The number of demonstrated current-time traces is limited and the sample ratio is not presented, i.e., it is not clarified whether it is an accidental behavior or a general reproducible phenomenon for the target molecule at the specific position.

Here are some detailed comments for the manuscripts:

We thank this reviewer for recommending the publication of our manuscript and for providing constructive comments and suggestions.

We addressed the individual concerns/questions from the reviewer as below.

1. I am wonder why the current show the telegram noise behavior when the Ag tip approaches the molecule. If the telegram noise behavior is due to the breaking and reforming of O-Si bonds at one end of the molecule upon SPP, then O-Si bonds at another end of the molecule can also break upon SPP, which will result in a three-states of current rather than two-level of current. Additionally, I am wonder why the O-Si bonds can reform frequently. If the break of O-Si is induced by the SPP enhanced vibration (Science, 2018, 360, 521), then, it will be great challenge to reform O-Si bonds frequently under a continuous resonant light illumination.

PTCDA has two anhydride groups at each end, both of which can dissociate from and reconnect to the surface under SPP. As the reviewer pointed out, this could lead to a three-level current switching: one from OFF configuration and two from the two different ON configurations resulting from the dissociation of each end. However, we did not find evidence for three-level switching for PTCDA, regardless of the tip location and tip height relative to the target molecule. Our explanation for this observation is as follows:

- (1) When the tip is located near the centre of the molecule, the difference in contact current between the two ON configurations may be too small to be detected in our measurements.
- (2) However, when the tip is positioned on one side of the molecule, the dissociation of the opposite end is energetically unfavourable, as it requires a much larger tilting angle to form a contact with the tip.
- (3) Both anhydride groups can be simultaneously dissociated by SPP excitation, leading to an irreversible change of the single-molecule junction. This is accompanied by the escape of the PTCDA molecule from the junction, as confirmed by subsequent STM images. This destructive operation was inevitable for PTCDA and was occasionally observed during our consecutive measurements. In this regard, as emphasized in the main text, the PMI molecule, with one imide end firmly bound to the surface, constitutes a more reliable and non-destructive switching junction.

The reason O–Si bonds frequently reform under illumination is that laser irradiation induces not only the forward reaction (OFF→ON) but also the reverse reaction (ON→OFF) which involves the dissociation of the tip–molecule interaction and the recovery of the molecule–surface bonds.

2. Please repeat the collection of current traces for this molecule at different locations to support the conclusion. Whether the switching behavior occurs every time and or once in a while. what is the percentage to observe such a behavior and what is the switching ratio. The above mentioned data can be statistically analyzed.

We thank the reviewer for his/hers suggestion and note that we did check carefully the reproducibility. We added the following sentence in the revised manuscript to address the reproducibility (Line 201);

Note that the switching event is not coincidental but well reproducible; once we found the proper tip conditions, we could repeatedly obtain switching features until the junction deformed (see also Supplementary Note 3 for details).

Furthermore, we supplemented a new note in the revised Supplementary Information (SI) to provide detailed information we could learn from the experiments regarding this issue;

Supplementary Note 3: Reproducibility of the switching event

Based on our repeated measurements, we are confident that the switching behaviour is a highly reproducible phenomenon. As statistical evidence, we found out of 115 trials, 101 cases showed switching, i.e switching occurs nearly every time. Each trial here consists of a sequence of tip approach and retraction on PTCDA under 532 nm irradiation (such as Fig. 2 in the main text or Supplementary Fig. 3), with data collected at both 78 and 10 K. We attribute the 14 non-switching cases (out of 115 trials) to one of the following scenarios, or the combinations of them: (1) the atomic structure of the tip apex being inappropriate for a sufficient attractive interaction with the molecule, (2) stronger molecule–surface bonds that do not dissociate under the same conditions, likely due to atomic-scale defects underneath the molecule that are not discernible with STM images, and/or (3) the tip height not reaching the switchable zone during the approach–retraction tip motion. Importantly, once we found the proper target molecule and tip conditions for switching, we could repeatedly obtain switching features until the junction deformed. As an example, we conducted 9 consecutive measurements (each consisting of a tip approach–retraction set) on the same molecule using the same tip, all of which demonstrated the switching features. This clearly indicates that the switching behaviour is not coincidental.

We confirm that the switching occurs at many different tip locations by collecting telegraphic traces of current and TERS patterns in Zone II with the tip precisely positioned within the molecules. The typical cases of the results are shown in Figure 3 for PMI and in Supplementary Figure 5 for PTCDA.

We did not find a specific correlation between the switching rate and the lateral tip position. As we have shown in the manuscript, the switching behaviour is much more susceptible to the vertical tip height.

3. Whether the molecule under test has light absorption at the laser wavelength used, it is recommended to complement the UV-Vis spectrum to exclude this factor.

According to the literature for the UV-Vis absorption spectrum for PTCDA nanostructures [R1], the molecule does not absorb light at 633 and 780 nm, where the switching behaviour was induced by SPP (Supplementary Figure 8 for 633 nm, Figure 4e for 780 nm). We note, that the UV-Vis spectrum of bulk PTCDA would not be directly relevant to the light absorption of single-molecule PTCDA on Si(111). Nevertheless, our dI/dV measurements shown in Supplementary Figure 7e provides evidence that resonant light absorption is not a primary contributor.

4. In Supplementary Note 2, it gives the reason for the decrease in TERS intensity as the tip gets closer to the surface in the on-state. It may possible due to the change of different plasmonic modes in the sub-nanometer quantum tunneling regime, from the BDP mode (charge transfer quenches the dipolar bonding plasmon (BDP), leading to decreasing plasmonic enhancement with the decreasing gap) to the CTP mode (the intensity of charge transfer plasmon (CTP) modes increases with the decreasing gap).

We appreciate the reviewer for highlighting a possibility to explain the decrease in TERS intensity at closer tip heights. On the one hand, we agree that quenching of the BDP mode in the tunnelling regime can contribute to this phenomenon at sub-nanometre distances. On the other hand, we note, that the relative intensities of each peak are changing for spectra at far and close tip height (the red and blue curves in Supplementary Fig. 2). This result cannot be explained by a reduction of the electromagnetic field in the junction but suggests a configurational change of the molecule at the two tip heights.

We have added the following sentence to address this possibility in Supplementary Note 2;

Moreover, the resonance-frequency shift and intensity decrease of the plasmon mode at shorter gap distances [4] may also contribute to the overall decrease in the TERS intensity.

5. During the movement of the tip, the on or off state of the current appears to be random rather than a regular telegraph behavior (Fig 2b). Can the authors really achieve controllable reversibility of the single-molecule switching? Does this mean that the breaking and forming of O-Si bonds is a random process?

For the measurements shown in Figure 2, we used relatively intense laser irradiation ($P_{\text{ext}} = 5.6 \text{ mW}$) to obtain sufficient TERS peak intensities. Under the conditions, the switching rate is high and the telegraph behaviour looks dense in the time trace. When reducing the laser power, we could clearly resolve the two-level telegraph signal in the current, as demonstrated in Supplementary Figure 4a-c ($P_{\text{ext}} = 8 \mu\text{W}$). By adjusting the tip height, we made subtle modification in relative energy between the OFF and ON states, thereby controlling the switching rate and the ON/OFF occupation in a reversible manner. We were able to fit the probability density of residence time for both ON and OFF states with a single exponential function (Supplementary Figure 4d-f), indicating that the switching process is stochastic.

6. The authors have performed laser power dependence tests (Figure S7). It shows that the

switching ratio is positively dependent on the laser power. While, this observation seems to conflict with the conclusions. When the laser power increase, the break of O-Si will become easy while the reformation of O-Si will become difficult simultaneously due to the asymmetric energy landscape upon SPP (Sci. Adv. 2022, 8, eabp9285).

The laser power dependence shown in Supplementary Figure 7d pertains to the OFF → ON transition rate, which is associated with the breaking of O–Si bonds. This shows that the bond breaking becomes easier under stronger laser illumination. The OFF → ON rate is derived from the inverse of the OFF state residence time at each laser power, which is basically unrelated to how frequently the reverse reaction (ON → OFF) occurs.

The residence time in the ON state strongly depends on the gap distance of the junction (Supplementary Figure 4). Changing the laser power induces variations in the gap distance at the junction due to thermal expansion, complicating direct comparisons of the ON → OFF rate at different laser powers. However, based on the observation that the OFF → ON and ON → OFF processes occur on similar timescales, we infer that the ON → OFF reaction is also triggered by SPP-generated hot carriers. Therefore, it is likely that the ON → OFF rate also increases with laser power.

7. Is it possible that the switching behavior is due to the action of optical gradient forces or the interaction between picocavity adatom dipole and the induced molecular dipole, which makes the bonds break or makes the tip atoms displacements (refer to Sci. Adv. 2022, 8, eabp9285).

We appreciate the reviewer's suggestion for possible mechanisms explaining the switching behaviour. However, we do not believe that the suggested mechanisms are predominant in our case and argue as follows: We compare measurements on the three types of molecules (Fig. 3). The absence of switching on the PDI molecule, despite its nearly identical adsorption geometry and central perylene structure shared with PTCDA and PMI, suggests that the switching is not due to tip atom displacement. Additionally, we do not expect that *unidirectional* optical gradient forces under continuous laser illumination at the junction can induce the *reversible* switching we observed.

The minor contribution of dipolar interactions between the molecules and picocavity adatom is inferred from the lack of significant distinction in switching behaviour between PMI and PTCDA under the same conditions. It is noteworthy that PMI has a large dipole moment (~3 Debye for an isolated molecule), whereas PTCDA does not.

To clarify the difference from the previous report, we added the following sentence in the revised manuscript (Line 294) along with the citation to Sci. Adv. 2022, 8, eabp9285;

The inertness of the imide groups strongly supports that the switching behaviour originates from the reactivity of the molecule–substrate system (Fig. 1a), ruling out the possibility of the switching due to the atomic-scale deformation of the plasmonic electrode (tip apex) in the picocavity reported previously [39].

8. For COMSOL based calculations, the distance of the gap is on the sub-nanometer scale, whether the finite element method is still applicable in this case and whether additional quantum effects need to be taken into account.

We agree that in sub-nanometre gaps, the classical approximation may not hold and quantum effects should be considered. On the other hand, a previous report [R2] using time-dependent density functional theory (TDDFT) and finite element method (FEM) with the quantum-corrected model (QCM) have

shown that significant quantum effects appear when the gap is less than 0.5 nm. Our FEM simulations in this manuscript were performed for wider gaps ($z > 0.5$ nm), where quantum effects are thus expected to be almost negligible.

To clarify it, we added the following sentence in the SI (Supplementary Note 7) of the revised manuscript:

Note that quantum effect correction [4] is not considered here, whereas it is required at shorter gap distances ($z < 0.5$ nm).

The FEM simulation of the Ag–Si heteromaterial nanojunction using QCM is a future challenge; in the previous report, only homomaterial gaps such as a Au–Au junction were considered [R2].

9. For discussion, can imide form N–Si bond on the substrate? Both N and O are electronegative, can other atoms be substituted here, such as C?

The N atom in the imide group does not form an N–Si bond because there is no Si atom directly beneath it. Similarly, the central O atom in the anhydride group also does not form a bond with the substrate. The difference in adsorption energy between imide and anhydride on the substrate is not due to the direct interaction of the central N or O atom with the surface. Instead, it is believed to originate from the mesomeric effect, where N acts as a better π -electron donor than O to the carbonyl C=O groups that directly bond with surface Si atoms.

Substitution with C at this site is possible by chemical synthesis [R3, R4]. However, it would lead to a transformation into a tautomerised form (*dihydroxyl-peropyrenequinone*), which is more stable than *tetra-carbonyl* form (see the scheme below). For this reason, we believe that the C-substituted molecule would not provide a direct comparison with PTCDA and PDI. Nevertheless, it could be an attractive candidate for separate investigation due to its potential for tautomerisation at the Si binding site.

10. The authors mentioned “Conventional molecular photoswitches, however, frequently rely on photochromic molecules, such as azobenzenes and piropyrans”, here, it is better to call it “azobenzene derivatives and piropyran derivatives” (J. Am. Chem. Soc. 2024, 146, 6856). Additionally, it is better to cite a newest review paper to give a comprehensive story for the auditors, e.g., Toward Practical Single-Molecule/Atom Switches (Adv. Sci. 2024, 2400877).

We corrected the sentence accordingly and added the two citations in the revised manuscript (Lines 68–71).

Reply to Comments of Reviewer 2

I am happy to recommend publication of this manuscript in its current form. The authors provide convincing data for the single molecule switching of PTCDA and its derivatives which is induced by the plasmonic field in the STM nanogap with a silver tip and a silicon surface. The STM imaging and associated STM and spectroscopic data is of good quality. Control experiments have been carefully conducted to strongly prove the hypothesis. The switching is well mapped out through the 3 different height zones, I, II and III and simultaneously recorded TERS spectra record unequivocal evidence. The 3 molecules compared (PTCDA, PMI and PDI) and the associated position resolved STM data provide an excellent comparison, and the role of the adsorption energy is well clarified. I also like the control experiments with the two different STM tip shapes with different plasmonic responses. These and the other control experiments help to make a clear and exacting assignment to plasmonic phenomena. This is an important new contribution to the field of nano-plasmonics and single molecule plasmonic phenomena.

We thank the reviewer for providing his/her positive comments and for recommending the publication of our manuscript.

Reply to Comments of Reviewer 3

The ms reports measurements of the tunnel current and TERS of PTCDA derivatives on silicon using a silver tip under irradiation. The authors observe two level fluctuations of the current as a function of the tip-substrate distance that they interpret as switching, i.e. uplifting the molecule to form a point contact with the tip. The authors investigate three different species with varying coupling strength to the substrate/tip and different positions of the tip on the molecule. Further control experiments are performed to support that plasmonic properties of the tip are instrumental for the observations. From the set of data the authors conclude that hot carriers mediate the switching. The data are in parts supported by DFT calculations.

The results are novel, interesting and in general well explained and should be published. I have a few comments though:

We thank the reviewer for recommending the publication of our manuscript and for acknowledging the novelty and important insights of this study.

From below we reply the individual comments/questions from the reviewer.

1. It is really necessary to use the term pico-optoelectronics? I find this unnecessary and a bit overselling. This report is by far not the first investigating light-induced molecular instabilities. The new aspect here is that it is on a semiconductor surface, but this does not justify to invent a new name for it. In addition the molecules are not particularly small, they are several nm large. The only picoscale is the lift height, which is again a typical height scale in STM business. Furthermore this term does not do anything to the science discussed here. I suggest to remove this term.

We acknowledge that the results of this manuscript do not fully conform to this terminology. In the revised manuscript, we removed this term from the abstract and introduction. However, we would like to emphasize that our control of the plasmon-driven molecular switch is much more precise than plasmon-induced reactions reported previously (e.g. Refs. 8 and 9 (9 and 10) in the previous (revised) manuscript). Even if a general STM can control the tip height in the 10 pm scale, it is not obvious that the 10-pm tip-height variation determines the reactivity of the target molecule. Therefore, we believe that this study well demonstrates the controllability of the plasmon-driven single-molecule devices *towards* pico-optoelectronics, and rewrote the last sentence of the conclusion in the revised manuscript as follows (Line 487);

The design and control of metal–molecule–semiconductor junctions should pave the way for the development of plasmon chemistry with high selectivity and controllability beyond conventional nano-optoelectronics, towards “pico-optoelectronics.”

2. The authors should define what they call a molecular switch. They use a wider definition than in many other works. In the present ms nothing switches inside the molecule. It is the contact to the tip that switches which is not an intrinsic property of the molecule under investigation.

We used the term “molecular switch” and “switching” for the process studied here, because the molecule can reversibly break and re-form chemical bonds with the surface, resulting in bi-stable conductance states by forming and breaking the contact with the tip. Although the molecule itself does not undergo isomerisation as in conventional molecular switches, the reversible changes in molecule–surface chemical bonding justify our use of the term. The molecular properties, such as the sharp contrast in

photoreactivity between anhydride and imide groups, determine the switching capability, which therefore reinforces our classification of this system as a molecular switch.

3. I find the quoted references not really adequately chosen. Several groups have investigated molecules on semiconductor surfaces and also "switching" behavior without always using this term, e.g. the Grill group, the Ernstorfer group and several others. Their work should also be referred to. Instead the authors refer extensively to their own work in a summarizing manner (e.g. refs. 22-25), where a reference to one single work would have been sufficient.

We followed the reviewer's suggestion and referenced reports from several groups including the Grill group regarding switching molecules on semiconductor surfaces such as Si, TiO₂, and GaAs [R5–R8]. However, we cannot find suitable reference from "the Ernstorfer group." Although we recognize that Professor Ralph Ernstorfer and his coworkers have studied fast dynamics of molecules on semiconductors, the methods and targets are far from those of this manuscript. Instead, we also cited topical reviews [R9, R10] on control of single atoms/molecules on silicon surfaces, covering works conducted by various research groups.

We added the following phrase along with the citations in the revised manuscript (Line 101):

...given the substantial variety of functional molecule–semiconductor systems [17–22],...

Regarding Refs. 22–25 of the original manuscript, the previous works reported by some of us, we removed Ref. 24 from the revised manuscript. The other references serve specific roles in our manuscript that cannot be replaced or removed. Ref. 22 provides detailed information on our experimental setup, including STM-TERS and FIB processed tip. Ref. 23 is crucial for assigning the Si bulk phonon peak observed in TERS. Ref. 25 is essential as it concerns point contact TERS on a molecule, which is directly relevant to this manuscript for determining the molecular configuration in the ON state based on TERS.

4. In line 168 the authors write: "Plasmon induced switching occurs when a plasmonic Ag tip approaches the surface...". The authors should add the information that this happens under irradiation and not just by approaching. The term "plasmonic tip" alone is not sufficiently well

The reviewer is correct and we thank the reviewer for pointing this out. We corrected the sentence accordingly in the revised manuscript (Line 157). Similar wording in Conclusion was also corrected appropriately (Line 466).

5. In line 217 the authors refer to quantised conductance to be seen in Fig. 2b. I do not see any quantized conductance there, nowhere in this work. Also the reference [25] is not an adequate reference for quantised conductance. A quantised conductance would give rise to currents in the range of 30 μA for 400 mV. The current here is in the nA range. Probably the authors mean discrete or saturated values of the conductance, but even these are hard to detect and that Fig. In Fig 1b one sees constant current over a certain period of time, but also no quantization. A constant current may also signal saturation of the amplifier. This issue must be resolved.

We apologize that we did not use the proper wording to describe the phenomenon. We intended to argue that the current became constant due to the point-contact formation by the tip approach. We have reworded the sentence appropriately as follows (Line 217):

The invariant conductance across different tip heights indicates the formation of a point contact between the tip and the molecule [34], consistent with the assignment by TERS described above.

6. It would also be fair to refer to other works in which simultaneously single-molecule Raman and conductance have been studied, e.g. by the Natelson group and the Barth group. established to understand that this happens under irradiation only.

We rewrote the corresponding sentence in the revised manuscript (Line 162) with the citations to the report from several groups including the Natelson group and the Barth group [R11–R14];

Single-molecule junctions can be further characterised by vibrational spectroscopy recorded simultaneously with conductance measurements [28–31].

References

- [R1] Han, Y. et al. Preparation, optical and electrical properties of PTCDA nanostructures. *Nanoscale* **7**, 17116–17121 (2015).
- [R2] Esteban, R., Borisov, A. G., Nordlander, P. & Aizpurua, J. Bridging quantum and classical plasmonics with a quantum-corrected model. *Nat. Commun.* **3**, 825 (2012).
- [R3] Buffet, N., Grelet, É. & Bock, H. Soluble and columnar liquid crystalline peropyrenequinones by coupling of phenalenones in caesium hydroxide. *Chem. Eur. J.* **16**, 5549–5553 (2010).
- [R4] Werner, S., Vollgraff, T. & Sundermeyer, J. Access to functionalized pyrenes, peropyrenes, terropyrenes, and quarterterpyrenes via reductive aromatization. *Angew. Chem. Int. Ed.* **60**, 13631–13635 (2021).
- [R5] Lastapis, M. et al., Picometer-scale electronic control of molecular dynamics inside a single molecule. *Science* **308**, 1000–1003 (2005).
- [R6] Vezzoli, A. et al., Single-molecule photocurrent at a metal–molecule–semiconductor junction. *Nano Lett.* **17**, 6702–6707 (2017).
- [R7] Rusimova, K. R. et al. Regulating the femtosecond excited-state lifetime of a single molecule. *Science* **361**, 1012–1016 (2018).
- [R8] Jacobson, P. et al. Adsorption and motion of single molecular motors on TiO₂(110). *J. Phys. Chem. C* **124**, 24776–24785 (2020).
- [R9] Leftwich, T. R. & Teplyakov, A. V., Chemical manipulation of multifunctional hydrocarbons on silicon surfaces. *Surf. Sci. Rep.* **63**, 1–71 (2008).
- [R10] Pitters, J. Et al. Atomically precise manufacturing of silicon electronics. *ACS Nano* **18**, 6766–6816 (2024).
- [R11] Ward, D.R et al., Simultaneous measurements of electronic conduction and Raman response in molecular junctions. *Nano Lett.* **8**, 919–924 (2008).
- [R12] Liu, Z. et al., Revealing the molecular structure of single-molecule junctions in different conductance states by fishing-mode tip-enhanced Raman spectroscopy, *Nat. Commun.* **2**, 305 (2011).
- [R13] Konishi, T. et al., Single molecule dynamics at a mechanically controllable break junction in solution at room temperature. *J. Am. Chem. Soc.* **135**, 1009–1014 (2013)
- [R14] Bi, H., et al., Voltage-driven conformational switching with distinct Raman signature in a single-molecule junction. *J. Am. Chem. Soc.* **140**, 4835–4840 (2018).

REVIEWERS' COMMENTS

Reviewer #1 (Remarks to the Author):

The interaction between SPP and single molecule is a very important research topic. The authors had revised the manuscript according to my suggestion. I would like to recommend to accept it.

Reviewer #3 (Remarks to the Author):

The ms. has adequately been revised. I agree with the authors that their functionality can be called a molecular switch. Still, I think it should be mentioned/explained that they use this term in a wider sense, since in their introduction they explicitly refer to those classes of photoswitchable molecules that do undergo a conformational change. They do not refer to those tons of papers which observed molecular switching by bond breaking/forming with the substrate (mostly on metallic substrates, admittedly).

As of the papers from the Ernstorfer group: It is correct that none of their papers is labeled "switching" (as I wrote in my first report). They call it "structural dynamics" and they do not aim at making switches, and it happens on faster time scales than reported in the present ms. Still, since the authors use this very wide definition of switching, they should give credit to those scientists who opened the field, i.e. studying the dynamics of molecules on surfaces with optical methods.

Reply to Comments of Reviewer 1

The interaction between SPP and single molecule is a very important research topic. The authors had revised the manuscript according to my suggestion. I would like to recommend to accept it.

We would like to thank the reviewer again for recommending the publication of our manuscript.

Reply to Comments of Reviewer 3

The ms. has adequately been revised. I agree with the authors that their functionality can be called a molecular switch. Still, I think it should be mentioned/explained that they use this term in a wider sense, since in their introduction they explicitly refer to those classes of photoswitchable molecules that do undergo a conformational change. They do not refer to those tons of papers which observed molecular switching by bond breaking/forming with the substrate (mostly on metallic substrates, admittedly).

Thank you for acknowledging the appropriateness of our revisions. In order to clarify the differences between our switch and the existing photoswitchable molecules and in order to define our observed behaviour as a switch using references of previous molecular switching with bond breaking/forming with the substrate, the following sentence was added in the revised manuscript (Line 128):

No intramolecular reaction occurs in PTCDA, unlike the conventional molecular photoswitches which operate through the reactions of photochromic moieties [5, 6]; nevertheless, the reversible formation and dissociation of the molecule–electrode interactions allow the system to be classified as a single-molecule switch [31–37].

As of the papers from the Ernstorfer group: It is correct that none of their papers is labeled "switching" (as I wrote in my first report). They call it "structural dynamics" and they do not aim at making switches, and it happens on faster time scales than reported in the present ms. Still, since the authors use this very wide definition of switching, they should give credit to those scientists who opened the field, i.e. studying the dynamics of molecules on surfaces with optical methods.

The following review paper on the dynamics studies for molecule–semiconductor systems pioneered by Prof. Ernstorfer and his coworkers was cited as Ref. 23 in the revised manuscript:

Gundlach, L., Ernstorfer, R., Willig, F.: Ultrafast interfacial electron transfer from the excited state of anchored molecules into a semiconductor. *Prog. Surf. Sci.* 82, 355–377 (2007).